# Identifying Principal Stratum Causal Effects Conditional on a Post-treatment Intermediate Response

**Xiaoqing Tan**                                                        XIT31@PITT.EDU
*University of Pittsburgh*

**Judah Abberbock**                                              ABBERBOCK@GMAIL.COM
*GlaxoSmithKline*

**Priya Rastogi**                                                  RASTOGIP@UPMC.EDU
*University of Pittsburgh*

**Gong Tang**                                                          GOT1@PITT.EDU
*University of Pittsburgh*

**Editors:** Bernhard Schölkopf, Caroline Uhler and Kun Zhang

## Abstract

In neoadjuvant trials on early-stage breast cancer, patients are usually randomized into a control group and a treatment group with an additional target therapy. Early efficacy of the new regimen is assessed via the binary pathological complete response (pCR) and the eventual efficacy is assessed via long-term clinical outcomes such as survival. Although pCR is strongly associated with survival, it has not been confirmed as a surrogate endpoint. To fully understand its clinical implication, it is important to establish causal estimands such as the causal effect in survival for patients who would achieve pCR under the new regimen. Under the principal stratification framework, previous studies focus on sensitivity analyses by varying model parameters in an imposed model on counterfactual outcomes. Under mild assumptions, we propose an approach to estimate those model parameters using empirical data and subsequently the causal estimand of interest. We also extend our approach to address censored outcome data. The proposed method is applied to a recent clinical trial and its performance is evaluated via simulation studies.

**Keywords:** Causal inference; Principal stratification; Identification; Randomized neoadjuvant trial; Censored outcome data.

## 1. Introduction

We have seen a major shift in the conduct of breast cancer clinical trials in recent years. Traditionally, breast cancer patients are randomly assigned to control or treatment after the primary surgery. Patients from the two groups are then followed over years for comparison of their long-term outcomes such as disease-free survival and overall survival. However, in recent years, there have been an increasing number of neoadjuvant trials where many of the systemic therapies are administered prior to the breast surgery (FDA, 2014).

The primary endpoint in neoadjuvant breast cancer clinical trials is pathological complete response (pCR), a binary indicator of absence of invasive cancer in the breast and auxiliary nodes (FDA, 2014). The rationale for using pCR is that efficacy of a treatment can be assessed at the time of surgery instead of the typical 5-10 years of follow-up on survival endpoints in the adjuvant setting. Strong association between pCR and survival has been well documented (Cortazar et al., 2014; Von Minckwitz et al., 2012), making pCR an attractive candidate surrogate. In the latest guidance of the U.S. Food and Drug administration (FDA), pCR is accepted as an endpoint to support

accelerated drug approvals, provided certain requirements are met (FDA, 2014). It is important to decipher the causal relationship among treatment, pCR, and survival in order to interpret the efficacy in survival when pCR is involved.

In the recently published National Surgical Adjuvant Breast and Bowel Project (NSABP) B-40 trial, patients with operable human epidermal growth factor receptor 2 (HER2)-negative breast cancer were randomly assigned to receive or not to receive bevacizumab along with their neoadjuvant chemotherapy regimens (Bear et al., 2012). The addition of bevacizumab significantly increased the rate of pCR. In terms of the long-term outcomes, patients on bevacizumab showed improvements in event-free survival (EFS) and overall survival (OS) compared to the control patients (Bear et al., 2015). Some investigators are interested in the comparison of survival between pCR patients in the treatment group and pCR patients in the control group. Such comparison, however, is problematic because these two groups of pCR patients are different and any direct comparison between them lacks causal interpretation.

Under the counterfactual framework (Rubin, 1974), potentially a patient has a pCR status after taking the control regimen and a pCR status after taking the treatment. Similarly, one can define counterfactual outcomes and causal effects in survival status (0/1) after a certain time period such as three years. The principal stratum framework proposed by Frangakis and Rubin (2002) can be used to describe causal effect in long-term outcomes (such as EFS) with an intermediate outcome (such as pCR) involved. Each principal stratum consists of subjects with the same pair of potential pCR status: the pCR status under the control regimen and the pCR status under the treatment regimen. One can then define the causal effect of treatment in EFS on each principal stratum.

Here we propose a method to identify and estimate principal stratum causal effects for a binary outcome and later extend our method for censored outcome data. The causal estimand of interest is the treatment efficacy in 3-year EFS and OS among patients who would achieve pCR under chemotherapy plus bevacizumab as in our motivating study, the NSABP B-40 trial. A model of counterfactual outcome given the observed data is imposed. Using some probabilistic arguments, we connect the model parameters with quantities that can be empirically estimated from the observed data. The resulting equations allow us to estimate the model parameters and subsequently the causal estimand of interest, and resolve the identifiability issue.

Our paper is organized as follows. Section 2 presents related work in principal stratum causal effects. Section 3 introduces the standard data settings, causal estimands of interest, and a regression model in the context of a randomized neoadjuvant trial. In Section 4, we provide key assumptions for identification of the causal estimand and introduce the proposed method. In Section 5, we conduct a simulation study to assess the performance of our method in terms of bias and coverage of bootstrap confidence intervals. In Section 6, we apply the proposed method to the motivating NSABP B-40 study. We conclude with a discussion of the proposed method and future work in Section 7.

## 2. Related Work

Frangakis and Rubin (2002) propose to split study population into principal strata. Each principal stratum is by definition independent of treatment assignment since it contains information on counterfactual, or potential outcomes rather than the observed outcome for a specific treatment assignment. One can then define treatment effects on each principal stratum. Additionally, any union of the basic principal strata would also be a valid principal stratum as it leads to comparisons among

a common set of individuals. Gilbert et al. (2015) show the principal stratification framework is useful for evaluating whether and how treatment effects differs across subgroups characterized by the intermediate variable, thus being firmly associated with the utility of the treatment marker.

Identification of principal stratum causal effects is in general difficult. A major challenge is that we do not observe the individual membership of principal stratum because of its counterfactual nature (Gilbert and Hudgens, 2008; Wolfson and Gilbert, 2010). Under the principal stratification framework, Gilbert et al. (2003) propose to perform sensitivity analyses by varying model parameters in an imposed parametric model for counterfactual outcomes. Shepherd et al. (2006) and Jemiai et al. (2007) extend this sensitivity analyses approach by including baseline covariates in the model. These sensitivity analyses can provide researchers with a range of causal estimates under different values of the sensitivity parameters. In reality, however, it is often unclear what the plausible values are for these sensitivity parameters and the selected combinations may not be exhaustive. Li et al. (2010) and Zigler and Belin (2012) use Bayesian approaches to model the joint distribution of the counterfactual intermediate outcomes and long-term outcomes and incorporate prior information regarding non-identifiable associations. The lack of identifiability, however, still exists and is reflected by the over-coverage of confidence intervals in their simulation studies.

Principal stratum causal effects with regards to outcomes truncated by death are not identifiable without further assumptions (Zhang and Rubin, 2003; Kurland et al., 2009; Lee et al., 2010). Tchetgen Tchetgen (2014) identify causal effects by borrowing information from post-treatment risk factors of the intermittent outcome and the causal estimand may vary according to the selected risk factors. Instrumental variables are also introduced to provide information on the unobserved principal strata and the justification of that exclusion restriction assumption is often challenging (Ding et al., 2011; Wang et al., 2017).

All the above methods either fall into sensitivity analyses or require exclusion restriction assumptions. In this paper, we propose a method to identify and estimate principal stratum causal effects under data settings as Shepherd et al. (2006) for a binary outcome and later extend our method to address issues of censored outcome data under mild assumptions. Identification of the causal effect is achieved with the bias minimal and the coverage probabilities close to the nominal levels.

## 3. The Principal Stratification Framework of Interest

### 3.1. Standard Setting for Neoadjuvant Studies

Consider a neoadjuvant breast cancer clinical trial where patients are randomized to two treatment groups. For subject $i = 1, 2, \ldots, n$, let $Z_i \in \{0, 1\}$ be the binary treatment assignment; $X_i \in \Gamma = \{0, 1, \ldots, K\}$ be a baseline discrete covariate. A continuous baseline variable $X_i$ such as clinical tumor size, would be grouped into $K + 1$ categories based on scientific knowledge. We will discuss extensions to the scenarios with a continuous $X_i$ in Section 7. Throughout this paper, we assume that the stable unit treatment value assumption (SUTVA) (Rubin, 1980) holds: the potential outcomes of any individual $i$ are unrelated to the treatment assignment of other individuals. Then we can denote $S_i(Z_i) \in \{0, 1\}$ as a binary post-randomization intermediate response such as the pCR status for subject $i$ under treatment $Z_i$ (possibly counterfactual). And denote $Y_i\{Z_i, S_i(Z_i)\} = Y_i(Z_i) \in \{0, 1\}$ as a binary long-term outcome of interest such as the EFS status at 3-year after study entry for subject $i$ under treatment $Z_i$ (possibly counterfactual). For individual $i$, $\{Z_i, X_i, S_i(Z_i), Y_i(Z_i)\}$ represents the observed data of treatment assignment,

baseline covariate, intermediate response and long-term outcome. If $Z_i = 0$, $\{S_i(0), Y_i(0)\}$ are observed and $\{S_i(1), Y_i(1)\}$ are counterfactual. If $Z_i = 1$, then $\{S_i(1), Y_i(1)\}$ are observed and $\{S_i(0), Y_i(0)\}$ are counterfactual. Thus for individual $i$, the complete counterfactual data would be $\{Z_i, X_i, S_i(0), S_i(1), Y_i(0), Y_i(1)\}$. Another important assumption is the monotonicity assumption: $S_i(0) \leq S_i(1)$ (Angrist et al., 1996), as in the motivating NSABP B-40 study, addition of bevacizumab led to improved pCR (Bear et al., 2012). We also assume for subject $i$, the treatment assignment $Z_i$ is independent of $X_i$ and the potential outcomes.

Under the principal stratification framework, denote the principal strata to be $E_{jk} = \{i : S_i(0) = j, S_i(1) = k\}$, $j, k = 0, 1$. The principal stratum causal effects of interest are

$$\theta_{jk} = \mathbb{E}\{Y_i(1) - Y_i(0)|i \in E_{jk}\}, \quad j, k = 0, 1.$$

Under the monotonicity assumption, the principal stratum $E_{10}$ is empty. In the NSABP B-40 study, we are interested in the causal effect in $E_{01} \cup E_{11}$, those who would achieve pCR had they been treated with chemotherapy plus bevacizumab:

$$\theta = \mathbb{E}\{Y_i(1) - Y_i(0)|i \in E_{+1} = E_{01} \cup E_{11}\} = \mathbb{E}\{Y_i(1) - Y_i(0)|S_i(1) = 1\}.$$

Other principal stratum causal effects such as $\theta_{jk}$ can be estimated using a similar approach as we outline in Section 4.

### 3.2. Modeling a Counterfactual Outcome

In order to estimate the principal stratum causal effects, Gilbert et al. (2003) propose to use a logistic regression model for $\Pr\{S_i(1) = 1|S_i(0) = 0, Y_i(0)\}$ as

$$\Pr\{S_i(1) = 1|S_i(0) = 0, Y_i(0)\} = \text{logit}^{-1}\{\beta_0 + \beta_1 Y_i(0)\}.$$

Shepherd et al. (2006) further extend the logistic regression by incorporating baseline covariates $X_i$ as

$$\begin{aligned}
\Pr\{S_i(1) = 1|S_i(0) = 0, Y_i(0), X_i = x\} &= \text{logit}^{-1}\{\beta_0 + \beta_1 Y_i(0) + \beta_2 x\} \\
&= \frac{exp\{\beta_0 + \beta_1 Y_i(0) + \beta_2 x\}}{1 + exp\{\beta_0 + \beta_1 Y_i(0) + \beta_2 x\}}.
\end{aligned} \tag{1}$$

Jemiai et al. (2007) consider a more general model framework:

$$\Pr\{S_i(1) = 1|S_i(0) = 0, Y_i(0), X_i = x\} = w[r(x) + g\{Y_i(0), x\}]$$

where $w(u) \equiv \{1 + exp(-u)\}^{-1}$ and $g(\cdot, \cdot)$ is a known function. In the case of Shepherd et al. (2006), $g(u, v) = \beta_1 u$ with $\beta_1$ known. Jemiai et al. (2007) show that under the monotonicity assumption, inference could be made on $\theta$ for any fixed function $g$ and sensitivity analyses could be performed by varying $g$.

## 4. The Proposed Method

### 4.1. Key Identification Assumptions

Identification of causal effects is achieved through two key assumptions. First, the monotonicity assumption: $S_i(0) \leq S_i(1)$ (Angrist et al., 1996). That is, a subject who responds under the

control would respond if given the treatment. This monotonicity assumption could prove valuable (Bartolucci and Grilli, 2011) and can be justified in many scenarios that the additional therapy would help to improve the response. In the motivating NSABP B-40 study, addition of bevacizumab led to improved pCR (Bear et al., 2012). Second, a parametric model is used to describe the counterfactual response under the treatment for a control non-respondent. Both the future long-term outcome and a baseline covariate are predictors in this parametric model. It is required that the level of the covariates is at least of the same dimension of model parameters and the imposed linearity assumption is critical to identify and estimate those regression parameters. We will elaborate the second assumption in Section 4.2.

## 4.2. Identification of Model Parameters and Causal Estimands

As mentioned in Shepherd et al. (2006) and will be described in Section 4.4, when the parameters of model (1) are identified, the causal estimands can be identified.

**Lemma 1** *For any $x \in \Gamma = \{0, 1, \ldots, K\}$ and $y \in \{0, 1\}$, let $a_x = \Pr\{S(1) = 1 | S(0) = 0, X = x\}$ and $b_{xy} = \Pr\{Y(0) = y | S(0) = 0, X = x\}$. Let $\mathbf{a} = (a_0, a_1, \ldots, a_K)^T$ and $\mathbf{b}_y = (b_{y0}, b_{y1}, \ldots, b_{yK})^T$. Define $h_x(\boldsymbol{\beta}, \mathbf{a}, \mathbf{b}_0, \mathbf{b}_1) = a_x - \sum_{y=0}^{1} b_{xy} \text{logit}^{-1}\{\beta_0 + \beta_1 y + \beta_2 x\}$, and $H(\boldsymbol{\beta}, \mathbf{a}, \mathbf{b}_0, \mathbf{b}_1) = \{h_0(\beta, \mathbf{a}, \mathbf{b}_0, \mathbf{b}_1), \ldots, h_K(\beta, \mathbf{a}, \mathbf{b}_0, \mathbf{b}_1)\}^T$.*
*If $\text{rank}\{\frac{\partial H(\boldsymbol{\beta}, \mathbf{a}, \mathbf{b}_0, \mathbf{b}_1)}{\partial \boldsymbol{\beta}}\} = 3$, within the neighborhood of $\boldsymbol{\beta}$ there is a unique solution $\boldsymbol{\beta} = \psi(\mathbf{a}, \mathbf{b}_0, \mathbf{b}_1)$ such that $H\{\psi(\mathbf{a}, \mathbf{b}_0, \mathbf{b}_1), \mathbf{a}, \mathbf{b}_0, \mathbf{b}_1\} = 0$.*

**Proof** For all $x \in \Gamma$, we have

$$a_x = \Pr\{S(1) = 1 | S(0) = 0, X = x\} = \sum_{y=0}^{1} \Pr\{S(1) = 1, Y(0) = y | S(0) = 0, X = x\}$$
$$= \sum_{y=0}^{1} \Pr\{Y(0) = y | S(0) = 0, X = x\} \Pr\{S(1) = 1 | Y(0) = y, S(0) = 0, X = x\}$$
$$= \sum_{y=0}^{1} b_{xy} \text{logit}^{-1}(\beta_0 + \beta_1 y + \beta_2 x).$$

Hence, $H(\boldsymbol{\beta}, \mathbf{a}, \mathbf{b}_0, \mathbf{b}_1) = 0$ and $H(\cdot)$ is a smooth function of $\boldsymbol{\beta}, \mathbf{a}, \mathbf{b}_0$, and $\mathbf{b}_1$. By invoking the implicit function theorem, when $\text{rank}(\frac{\partial H}{\partial \boldsymbol{\beta}}) = 3$, there exists a smooth function $\psi$ such that $\boldsymbol{\beta} = \psi(\mathbf{a}, \mathbf{b}_0, \mathbf{b}_1)$ and $H\{\psi(\mathbf{a}, \mathbf{b}_0, \mathbf{b}_1), \mathbf{a}, \mathbf{b}_0, \mathbf{b}_1\} = 0$. ∎

The identifiability of model parameter $\boldsymbol{\beta}$ depends on the availability of $a_x = \Pr\{S(1) = 1 | S(0) = 0, X = x\}$ and $b_{xy} = \Pr\{Y(0) = y | S(0) = 0, X = x\}$, for $x \in \Gamma; y = 0, 1$. The linearity in $X = x$ in model (1) also plays an important role. In general, when $\beta_2 \neq 0$ and $K \geq 2$, there are equal or more equations than the number of unknown parameters in $\boldsymbol{\beta}$, Lemma 1 would hold. In practice, given $(\mathbf{a}, \mathbf{b}_0, \mathbf{b}_1)$, one solves for $\boldsymbol{\beta}$ such that $H(\boldsymbol{\beta}, \mathbf{a}, \mathbf{b}_0, \mathbf{b}_1) = 0$. Then verify that $\text{rank}\{\frac{\partial H(\boldsymbol{\beta}, \mathbf{a}, \mathbf{b}_0, \mathbf{b}_1)}{\partial \boldsymbol{\beta}}\} = 3$ at the solution.

## 4.3. Estimation of Causal Estimands

The causal estimand of interest is

$$\theta = \mathbb{E}\{Y_i(1) - Y_i(0) | S_i(1) = 1\} = \mathbb{E}\{Y_i(1) | S_i(1) = 1\} - \mathbb{E}\{Y_i(0) | S_i(1) = 1\}. \tag{2}$$

Because $\{Y_i(1), S_i(1)\}$ are observed for subjects in the treatment arm, $\Pr\{Y_i(1) = 1|S_i(1) = 1\}$ can be estimated by

$$\widehat{\Pr}\{Y_i(1) = 1|S_i(1) = 1\} = \frac{\sum_i I\{Z_i = 1, S_i(1) = 1, Y_i(1) = 1\}}{\sum_i I\{Z_i = 1, S_i(1) = 1\}}. \tag{3}$$

where $I(\cdot)$ is the indicator function.

Meanwhile,

$$\begin{aligned} \Pr\{Y_i(0) = 1|S_i(1) = 1\} &= \frac{\Pr\{S_i(1) = 1, Y_i(0) = 1\}}{\Pr\{S_i(1) = 1\}} \\ &= \frac{\sum_x \Pr\{S_i(1) = 1, Y_i(0) = 1|X_i = x\} \cdot \Pr\{X_i = x\}}{\sum_x \Pr\{S_i(1) = 1|X_i = x\} \cdot \Pr\{X_i = x\}} \end{aligned} \tag{4}$$

In equation (4), $\Pr\{X_i = x\}$ can be estimated by $\widehat{\Pr}\{X_i = x\} = \dfrac{\sum_i I(X_i = x)}{n}$ and

$$\begin{aligned} &\Pr\{S_i(1) = 1, Y_i(0) = 1|X_i = x\} \\ &= \sum_{j=0}^1 \Pr\{S_i(1) = 1, Y_i(0) = 1, S_i(0) = j|X_i = x\} \\ &= \sum_{j=0}^1 \Pr\{S_i(1) = 1, Y_i(0) = 1|S_i(0) = j, X_i = x\} \cdot \Pr\{S_i(0) = j|X_i = x\} \\ &= \sum_{j=0}^1 \Big[ \Pr\{S_i(1) = 1|S_i(0) = j, Y_i(0) = 1, X_i = x\} \\ &\quad \cdot \Pr\{Y_i(0) = 1|S_i(0) = j, X_i = x\} \cdot \Pr\{S_i(0) = j|X_i = x\} \Big]. \end{aligned} \tag{5}$$

In equation (5), $\Pr\{Y_i(0) = 1|S_i(0) = j, X_i = x\}$, $j = 0, 1$, can be estimated by

$$\widehat{\Pr}\{Y_i(0) = 1|S_i(0) = j, X_i = x\} = \frac{\sum_i I\{Z_i = 0, S_i(0) = j, Y_i(0) = 1, X_i = x\}}{\sum_i I\{Z_i = 0, S_i(0) = j, X_i = x\}}.$$

By the monotonicity assumption, $\Pr\{S_i(1) = 1|S_i(0) = 1, Y_i(0) = 1, X_i = x\} \equiv 1$.
The estimation of $\Pr\{S_i(j) = 1|X_i = x\}$, $j = 0, 1$, is described in Lemma 2.

**Lemma 2** *Under the monotonicity assumption, for any $x$, we denote*

$$\widehat{q}_j(x) = \frac{\sum_i I\{Z_i = j, S_i(j) = 1, X_i = x\}}{\sum_i I\{Z_i = j, X_i = x\}}, \ \ j = 0, 1;$$

*the observed proportions of responders in the control group and the treatment group with $X = x$, respectively.*

*We use maximum likelihood estimation to estimate $\Pr\{S_i(j) = 1|X_i = x\}$, $j = 0, 1$.*

*(a) when $\widehat{q}_1(x) \geq \widehat{q}_0(x)$, the maximum likelihood estimate of $\Pr\{S_i(j) = 1|X_i = x\}$ is $\widehat{q}_j(x)$, $j = 0, 1$;*

*(b) when $\widehat{q}_1(x) < \widehat{q}_0(x)$, the maximum likelihood estimate of $\Pr\{S_i(j) = 1|X_i = x\}$ is* $\dfrac{\sum_i I(S_i = 1, X_i = x)}{\sum_i I(X_i = x)}$, $j = 0, 1$.

In the second scenario, the estimates are the same as the pooled proportion of responders among patients with $X = x$. The proof of Lemma 2 is presented in Appendix A.

The last item in equation (4) needed for estimating the causal estimand is $\Pr\{S_i(1) = 1 | S_i(0) = 0, Y_i(0) = 1, X_i = x\}$. Gilbert et al. (2003) and Shepherd et al. (2006) conduct sensitivity analyses by varying the values of the $\boldsymbol{\beta}$ in model (1). In Section 4.4, we will discuss how to estimate $\boldsymbol{\beta}$ using a probabilistic equation.

### 4.4. Estimation of Model Parameters

Let

$$G_L(x) = \Pr\{S_i(1) = 1 | S_i(0) = 0, X_i = x\}$$
$$G_R(x, y) = \Pr\{Y_i(0) = y | S_i(0) = 0, X_i = x\}$$
$$G_M(x, y; \boldsymbol{\beta}) = \Pr\{S_i(1) = 1 | S_i(0) = 0, Y_i(0) = y, X_i = x\}.$$

This leads to an equation system:

$$G_L(x) = \sum_{y=0}^{1} G_M(x, y; \boldsymbol{\beta}) \cdot G_R(x, y); x \in \Gamma$$

We can estimate $G_L(x)$ with the following empirical estimates from the observed data by

$$\widehat{G}_L(x) = \frac{\widehat{\Pr}\{S_i(0) = 0, S_i(1) = 1 | X_i = x\}}{\widehat{\Pr}\{S_i(0) = 0 | X_i = x\}}$$

where the numerator and the denominator are derived from Lemma 2. The details are presented in Appendix A.

Because $\{X_i, S_i(0), Y_i(0)\}$ are observed for subjects in the control arm, $G_R(x, y)$ can be estimated by

$$\widehat{G}_R(x, y) = \frac{\sum_i I\{Z_i = 0, S_i(0) = 0, Y_i(0) = y, X_i = x\}}{\sum_i I\{Z_i = 0, S_i(0) = 0, X_i = x\}}$$

With $\widehat{G}_L(x)$ and $\widehat{G}_R(x, y)$ estimated from the observed data and $G_M(x, y; \boldsymbol{\beta})$ specified as the regression model in equation (1), we have

$$\widehat{G}_L(x) = \sum_{y=0}^{1} G_M(x, y; \boldsymbol{\beta}) \cdot \widehat{G}_R(x, y); \quad x \in \Gamma \tag{6}$$

The number of unknown parameters $\boldsymbol{\beta}$ in system of equations (6) is three and the number of equations is $(K + 1)$, for $X_i \in \Gamma = \{0, 1, \ldots, K\}$. For (6), when $K + 1 < 3$, we cannot uniquely solve for $\boldsymbol{\beta}$. When $K + 1 = 3$, the number of equations is the same as the number of unknown parameters and in general we can solve for $\boldsymbol{\beta}$. When $K + 1 > 3$, there are more equations than the number of unknown parameters, and there are generally no exact solutions to the equation systems (6). In that case, we propose to estimate $\boldsymbol{\beta}$ by

$$\widehat{\boldsymbol{\beta}} = \arg\min_{\boldsymbol{\beta}} \sum_{x=0}^{K} \{\widehat{G}_L(x) - \sum_{y=0}^{1} G_M(x, y; \boldsymbol{\beta}) \cdot \widehat{G}_R(x, y)\}^2 \tag{7}$$

where $\widehat{G}_L(x)$, $\widehat{G}_R(x, y)$ and $G_M(x, y; \boldsymbol{\beta})$ are probabilities bounded between 0 and 1.

With $\boldsymbol{\beta}$ estimated, we can estimate the causal estimand $\theta$ via the procedure outlined in Section 4.3.

### 4.5. Consistency of Model Parameters and Causal Estimands

Here we provide the theoretical guarantee of our estimators $\boldsymbol{\beta}$ and $\theta$.

Let

$$Q_0^{(x)}(\boldsymbol{\beta}) = \{G_L(x) - \sum_{y=0}^{1} G_M(x, y; \boldsymbol{\beta}) \cdot G_R(x, y)\}^2, \quad x \in \Gamma; y = 0, 1$$

$$\tilde{Q}_0(\boldsymbol{\beta}) = \{Q_0^{(0)}(\boldsymbol{\beta}), Q_0^{(1)}(\boldsymbol{\beta}), \ldots, Q_0^{(K)}(\boldsymbol{\beta})\}^T,$$

$$Q_n(\boldsymbol{\beta}) = \sum_{x=0}^{K} Q_n^{(x)}(\boldsymbol{\beta}) = \sum_{x=0}^{K} \{\widehat{G}_L(x) - \sum_{y=0}^{1} G_M(x, y; \boldsymbol{\beta}) \cdot \widehat{G}_R(x, y)\}^2$$

**Theorem 3** *Under the following conditions:*

*(a) $\boldsymbol{\beta}$ satisfies $Q_0^{(x)}(\boldsymbol{\beta}) = 0$, $\forall x \in \Gamma = \{0, 1, \ldots, K\}$.*

*(b) $\mathrm{rank} \left| \dfrac{\partial \tilde{Q}_0(\boldsymbol{\beta})}{\partial \boldsymbol{\beta}} \right| \geq \dim(\boldsymbol{\beta})$.*

*(c) $\widehat{G}_L(x) \xrightarrow{p} G_L(x), \widehat{G}_R(x, y) \xrightarrow{p} G_R(x, y)$, as $n \to \infty$, $\forall x \in \Gamma; \forall y = 0, 1$.*

*Then $\widehat{\boldsymbol{\beta}} = \arg \min_{\boldsymbol{\beta}} Q_n(\boldsymbol{\beta}) \xrightarrow{p} \boldsymbol{\beta}$ and the causal estimand $\widehat{\theta} \xrightarrow{p} \theta$ as $n \to \infty$.*

The detailed proof of Theorem 3 is presented in Appendix B.

### 4.6. Extension to Censored Data

As in the motivating NSABP B-40 study, the long-term outcome $Y_i$ may be subject to right censoring. For any time $T = t_0$ of interest, the binary counterfactual outcomes would be $\{Y_i(0; t_0), Y_i(1; t_0)\}$ and the causal estimand can be formulated as

$$\theta(t_0) = \mathbb{E}\{Y_i(1; t_0) - Y_i(0; t_0) | i \in E_{+1}\}.$$

With $Y_i$ subject to censoring, $\Pr\{Y_i(1; t_0) = 1 | i \in E_{+1}\}$ can be estimated by the Kaplan-Meier (KM) estimates at time $T = t_0$. The estimation is similar for other relevant quantities such as $\Pr\{Y_i(0; t_0) = 1 | S_i(0) = j, X_i = x\}$ in equation (5) under the scenario where $Y_i(Z_i)$ is always observed.

## 5. Simulation Studies

A simulation study is used to assess the performance of the proposed method. The setup is chosen to resemble the NSABP B-40 study by simulating treatment assignment, baseline tumor size category, binary pCR response status, and binary survival status, specifically:

$$\mathfrak{D} = [D_i = \{Z_i, X_i, S_i(0), S_i(1), Y_i(0), Y_i(1)\}, \ i = 1, \ldots, n].$$

We simulate the subject-level data as follows. First, we simulate the categorical baseline tumor category $X_i$ from a multinomial distribution with $\Pr\{X_i = x\} = 0.25, x \in \{0, 1, 2, 3\}$. Next, we simulate $S_i(0)$ given $X_i$ from a Bernoulli distribution with $\Pr\{S_i(0) = 1|X_i = x\} = p(x)$ with $p(0), p(1), p(2), p(3) = 0.3, 0.25, 0.25, 0.2$, respectively. We then simulate the survival status under control, $Y_i(0)$, with a Bernoulli draw with $\Pr\{Y_i(0) = 1|S_i(0) = 0, X_i = x\} = 0.7, 0.65, 0.6, 0.55$ for $x = 0, 1, 2, 3$, respectively and $\Pr\{Y_i(0) = 1|S_i(0) = 1, X_i = x\} = 0.84, 0.78, 0.72, 0.66$ for $x = 0, 1, 2, 3$, respectively. The choice of these numbers reflects a 20% improvement in 3-year EFS for respondents over nonrespondents under the control regimen.

Next, we simulate the conditional distribution $\{S_i(1)|S_i(0), Y_i(0), X_i\}$. For subjects with $S_i(0) = 1$ we set $S_i(1)$ to be 1 to enforce the monotonicity assumption. For subjects with $S_i(0) = 0$ we draw $S_i(1)$ from a Bernoulli distribution: $\Pr\{S_i(1) = 1|S_i(0) = 0, Y_i(0) = y, X_i = x\} = \dfrac{\exp(\beta_0 + \beta_1 y + \beta_2 x)}{1 + \exp(\beta_0 + \beta_1 y + \beta_2 x)}$. We try different settings for $\boldsymbol{\beta}$ = (-3, -5, 0.2), (-5, -1, -2), and (-7, 3, 0.2).

We then simulate the survival status under treatment, $Y_i(1)$, according to the following probability distributions:

$$\Pr\{Y_i(1) = 1|S_i(0) = 0, S_i(1) = 0, Y_i(0) = 0\} = 0.5,$$
$$\Pr\{Y_i(1) = 1|S_i(0) = 0, S_i(1) = 0, Y_i(0) = 1\} = 0.6,$$
$$\Pr\{Y_i(1) = 1|S_i(0) = 0, S_i(1) = 1, Y_i(0) = 0\} = 0.85,$$
$$\Pr\{Y_i(1) = 1|S_i(0) = 0, S_i(1) = 1, Y_i(0) = 1\} = 0.9,$$
$$\Pr\{Y_i(1) = 1|S_i(0) = 1, S_i(1) = 1, Y_i(0) = 0\} = 0.85,$$
$$\Pr\{Y_i(1) = 1|S_i(0) = 1, S_i(1) = 1, Y_i(0) = 1\} = 0.9.$$

These probabilities are chosen to make the 3-year EFS under treatment greater for those who would obtain pCR under treatment than those who would not, and have a greater 3-year EFS for those patients who would be event-free under control than those who would not be event-free under control. We set these probabilities to be independent of the baseline tumor size given the potential outcomes $\{S_i(0), S_i(1), Y_i(0)\}$.

Lastly we simulate the treatment assignment with equal probability for each arm as a Bernoulli draw with $\Pr\{Z_i = 0\}$ and $\Pr\{Z_i = 1\}$ both equal to 0.5 to ensure that independence between potential outcomes and treatment assignment. For the simulated data the true average causal effect for principal stratum $S_i(1) = 1$, $\mathbb{E}\{Y_i(1) - Y_i(0)|S_i(1) = 1\}$, can be calculated using the above parameters for simulations. The detailed calculations is given in Appendix C. Under the three parameter settings the true values of the causal estimands are $\theta$=0.179, 0.130, and 0.120, respectively. This means that under the three different settings, if the treatment was administered to all subjects who would achieve pCR under treatment there would be a 17.9%, 13.0%, 12.0% increment in survival respectively, within the time frame under consideration, than had all of them taken the control instead.

Under each parameter setting and a chosen sample size $n$=1000, 2000, or 4000, we simulate $R$=1000 replicates. A quasi-Newton method, the Broyden-Fletcher-Goldfarb-Shanno algorithm, is used for the optimization. We create $B$=500 bootstrap samples to obtain the 95% confidence interval for the causal estimates. Let $\widehat{\theta}^{(r)}$ be the mean estimate among bootstrap samples from the $r$ replicate, $r = 1, \ldots, R$.

We construct bootstrap confidence intervals to account for the variability introduced by estimating model parameters. We use the basic bootstrap CI, or the pivotal CI (Davison and Hinkley, 1997) for constructing CIs from bootstrap estimates. Let $\{\widehat{\theta}^{(1)}, \widehat{\theta}^{(2)}, \ldots, \widehat{\theta}^{(B)}\}$ are the causal effect estimates from $B$ bootstrap samples. Denote $\theta^*_{(1-\alpha/2)}$ and $\theta^*_{(\alpha/2)}$ as the $100(1-\alpha/2)\%$ and $100(\alpha/2)\%$ of the bootstrap causal effect estimates. The $100(1-\alpha)\%$ bootstrap confidence interval is given by $(2\widehat{\theta} - \theta^*_{(1-\alpha/2)}, 2\widehat{\theta} - \theta^*_{(\alpha/2)})$ where $\widehat{\theta}$ is the estimate from the data.

We report the empirical bias, mean squared error (MSE), average length of 95% CIs, and the coverage of those CIs. $\mathrm{Bias}(\widehat{\theta}) = \frac{1}{R}\sum_{r=1}^{R}\{\widehat{\theta}^{(r)} - \theta\}$, $\mathrm{MSE}(\widehat{\theta}) = \frac{1}{R}\sum_{r=1}^{R}\{\widehat{\theta}^{(r)} - \theta\}^2$, 95% CI width $= \frac{1}{R}\sum_{r=1}^{R}|\widehat{\theta}^{(r)}_{U,0.05} - \widehat{\theta}^{(r)}_{L,0.05}|$, and 95% CI coverage $= \frac{1}{R}\sum_{r=1}^{R}I\{\theta \in (\widehat{\theta}^{(r)}_{L,0.05}, \widehat{\theta}^{(r)}_{U,0.05})\}$ with $\widehat{\theta}^{(r)}_{L,0.05}$ and $\widehat{\theta}^{(r)}_{U,0.05}$ the lower bound and upper bound of the 95% bootstrap CIs of $\widehat{\theta}$ from the $r^{th}$ simulated dataset. Table 1 shows the simulation results of the proposed method under three different parameter settings and various sample sizes. Our simulation results show the identification of causal effects is achieved with the bias negligible and the coverage probabilities close to the nominal levels.

Table 1: Simulation results of the proposed method under three different parameter settings and various sample sizes.

| Sample size | Empirical Bias | MSE | 95% CI width | 95% CI coverage |
|---|---|---|---|---|
| Setting 1: $\boldsymbol{\beta}$=(-3, -5, 0.2), $\theta$=0.179 | | | | |
| 1000 | -0.011 | 3.001e-3 | 0.206 | 0.952 |
| 2000 | -0.006 | 1.539e-3 | 0.155 | 0.955 |
| 4000 | -0.002 | 6.755e-4 | 0.116 | 0.962 |
| Setting 2: $\boldsymbol{\beta}$=(-5, -1, -2), $\theta$=0.130 | | | | |
| 1000 | -6.011e-5 | 2.496e-3 | 0.185 | 0.943 |
| 2000 | 9.358e-4 | 1.137e-3 | 0.130 | 0.948 |
| 4000 | 1.086e-4 | 5.462e-4 | 0.093 | 0.950 |
| Setting 3: $\boldsymbol{\beta}$=(-7, 3, 0.2), $\theta$=0.120 | | | | |
| 1000 | 0.008 | 2.547e-3 | 0.194 | 0.955 |
| 2000 | 0.006 | 1.319e-3 | 0.141 | 0.957 |
| 4000 | 0.003 | 6.363e-4 | 0.100 | 0.953 |

## 6. Application to NSABP B-40 Trial

### 6.1. B-40 Data Analysis

Here we apply the proposed method to the NSABP B-40 study (Bear et al., 2012, 2015). Among the 1206 enrolled participants, 13 withdrew consent, 7 had missing data and 2 had had inoperable disease after chemotherapy. Another 15 patients did not have nodal assessment so their pCR status was not ascertained. We conduct our analysis among the rest 1169 patients. Our purpose is to estimate the causal treatment effect in 3-year EFS and OS among patients who would obtain a pCR

had bevacizumab been added to their treatment regimen. KM estimates are used since there are 61 patients censored at 3 years.

To apply our method, the clinical tumor size is used as the baseline auxiliary covariate $X$. Patients are grouped into four nearly equal-sized groups: 2-3 cm, 3.1-4 cm, 4.1-6 cm and >6 cm, based on breast cancer expert knowledge. We code these four tumor size groups into $\{0, 1, 2, 3\}$, respectively. Among the 589 patients in the control arm, the proportions of those who achieved pCR in each patient group are 28%, 23%, 22% and 17%, respectively; among the 580 patients in the treatment arm, the proportions of those who achieved pCR are 31%, 26%, 25% and 27%, respectively. This does not violate the monotonicity assumption $S_i(0) \leq S_i(1)$. The 3-year long-term outcome status $Y_i = 1$ if the patient $i$ survived within the first 3 years and 0 otherwise.

We calculate the 95% bootstrap confidence intervals from 500 bootstrap samples. The estimated causal treatment effect in 3-year EFS among those who would obtained pCR under treatment is $\widehat{\theta}_{\text{EFS}} = 0.180$ (95% CI=(0.056, 0.377)) with $\widehat{\beta} = (-1.797, -5.874, 0.285)$. The estimated causal treatment effect in 3-year OS among those who would obtained pCR under treatment is $\widehat{\theta}_{\text{OS}} = 0.175$ (95% CI=(0.062, 0.354)) with $\widehat{\beta} = (-1.85, -4.764, 0.289)$. For both scenarios, because 0 is outside of the 95% CIs, we would claim that the addition of bevacizumab improves 3-year EFS and OS among patients who would respond to neoadjuvant chemotherapy plus bevacizumab at a 95% confidence level.

## 6.2. Sensitivity of Initial Parameters in Optimization

For the real data application, the initial estimate $\boldsymbol{\beta}_{init} = (\beta_0, \beta_1, \beta_2)$ is set at $(0, 0, 0)$. To see the sensitivity of initial parameters, we try $9261 = 21 \times 21 \times 21$ different initial values of $\boldsymbol{\beta}_{init}$, with $\beta_0$, $\beta_1$, and $\beta_2$ on the integer grids of $[-10, 10] \times [-10, 10] \times [-10, 10]$. The corresponding histograms of causal estimates in 3-year EFS and 3-year OS at convergence are presented in Figure 1. Our estimated model parameters $\widehat{\boldsymbol{\beta}}$ in Section 6.1 achieves the minimum loss of equation (7). Except for some extreme initialization such as (10,10,10), most of the $\widehat{\theta}$ are the same or very close to the causal estimates calculated by using $\boldsymbol{\beta}_{init} = (0,0,0)$ as initial parameters. Therefore, we conclude that the causal estimand is not sensitive to the initial parameter settings in optimization. In practice, we suggest running optimization with various initial values and identify the right estimate.

## 6.3. Comparisons to Sensitivity Analysis Method

We compare the performance of our method with that of the sensitivity analysis similar to Gilbert et al. (2003) and Shepherd et al. (2006). Recall that for $X = x \in \Gamma = \{0, 1, \ldots, K\}$, we have an equation system:

$$\widehat{G}_L(x) = \sum_{y=0}^{1} G_M(x, y; \boldsymbol{\beta}) \cdot \widehat{G}_R(x, y); \quad x \in \Gamma = \{0, 1, \ldots, K\}$$

where $G_M(x, y; \boldsymbol{\beta}) = \dfrac{\exp(\beta_0 + \beta_1 y + \beta_2 x)}{1 + \exp(\beta_0 + \beta_1 y + \beta_2 x)}$. In the sensitivity analysis we vary the value of $\beta_1$ from -7 to -3. Then for each category of $x$ we define $\beta_x = \beta_0 + \beta_2 x$. Under this reparameterization we have only one unknown parameter, $\beta_x$, for each equation. We then solve for $\beta_x$ for each equation independently and obtain the causal estimand subsequently.

By varying values of $\beta_1$ around the estimated $\widehat{\beta}_1$ from Section 6.1, the corresponding causal estimands in 3-year EFS and 3-year OS are presented in Table 2. The estimated causal effects in

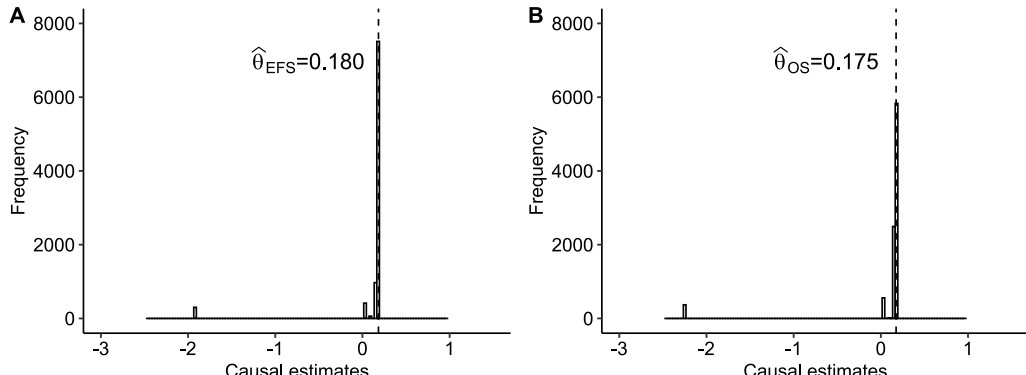

Figure 1: Histogram of the causal estimates $\widehat{\theta}$ obtained from $9261 = 21 \times 21 \times 21$ different initial values of $\boldsymbol{\beta}_{init}$ in the optimization process for 3-year EFS (Figure A) and 3-year OS (Figure B), respectively. The values of $\beta_0$, $\beta_1$ and $\beta_2$ vary on the integer grids of $[-10, 10] \times [-10, 10] \times [-10, 10]$. Except for some extreme initialization such as $\boldsymbol{\beta}_{init} = (10,10,10)$, most of the $\widehat{\theta}$ are the same or very close to the causal estimate calculated by using $\boldsymbol{\beta}_{init} = (0,0,0)$ as initial parameters.

3-year EFS vary from 0.159 to 0.181 with none of the 95% CIs including 0; the estimated causal effects in 3-year OS vary from 0.132 to 0.176 with none of the 95% CIs including 0. These intervals overlap a lot with the confidence intervals of real data. These results suggest the addition of bevacizumab may improve 3-year EFS and 3-year OS among patients who would respond to neoadjuvant chemotherapy plus bevacizumab.

Table 2: Sensitivity analysis for the estimated causal effect of bevacizumab in 3-year survival among those who would obtain pCR under chemotherapy plus bevacizumab.

| Long-term survival | $\beta_1$ | $\widehat{\theta}$ | 95% CI for $\widehat{\theta}$ |
|---|---|---|---|
| EFS | -7 | 0.181 | (0.025, 0.290) |
|  | -6 | 0.180 | (0.043, 0.289) |
|  | -5 | 0.178 | (0.040, 0.282) |
|  | -4 | 0.172 | (0.058, 0.272) |
|  | -3 | 0.159 | (0.065, 0.267) |
| OS | -7 | 0.176 | (0.055, 0.278) |
|  | -6 | 0.172 | (0.067, 0.267) |
|  | -5 | 0.166 | (0.069, 0.267) |
|  | -4 | 0.153 | (0.066, 0.235) |
|  | -3 | 0.132 | (0.064, 0.200) |

## 7. Discussion and Future Work

We have proposed a method under the principal stratification framework to estimate causal effects of a treatment on a binary long-term endpoint conditional on a post-treatment binary marker in randomized controlled clinical trials. We also extend our method to address censored outcome data. In our motivating study, we demonstrate the causal effect of the new regimen in the long-term survival for patients who would achieve pCR. Other principal stratum causal effects can be estimated in a similar fashion. Our approach can play an important role in a sensitivity analysis.

Identification of causal effects is achieved through two assumptions. First, a subject who responds under the control would respond if given the treatment. This monotonicity assumption could prove valuable (Bartolucci and Grilli, 2011) and can be justified in many scenarios that the additional therapy would help to improve the response. When the auxiliary variable $X$ is discrete, we can identify and estimate $\Pr\{S(1) = 1|S(0) = 0, X\}$ under the monotonicity assumption. Second, a parametric model is used to describe the counterfactual response under the treatment for a control non-respondent (Shepherd et al., 2006). Both the future long-term outcome and a baseline covariate are predictors in this parametric model. Shepherd et al. (2006) does not consider when the auxiliary $X$ is discrete, the parameters of model (1) can be identified when the level of the discrete covariate is at least of the same dimension of model parameters. Instead they perform sensitivity analyses by varying the values of those model parameters in order to estimate the causal estimands. It is recognized that no diagnostic tool is available to verify the validity of this counterfactual model.

In the motivating dataset, we discretize a continuous baseline variable into several levels. In practice, the linearity assumption may not hold. We would consider a two-pronged approach: 1) to estimate $G_L(x)$ and $G_R(x, y)$ by nonparametric estimates such as spline or kernel density estimates for a univariate continuous $X$; 2) to use a more flexible model for the counterfactual response such as a logistic regression with natural cubic spline with fixed and even-spaced knots along the domain of $X$. For each given $x$, we can still use the same probabilistic argument to link those estimates and the model parameters. The objective function would be a weighted sum of the squared difference of those probabilistic estimates.

## Acknowledgments

This work is supported by the National Cancer Institute at the National Institutes of Health, U.S. Department of Health and Human Services, Public Health Service grants U10-CA180868 (NCTN), U10-CA180822 (NRG SDMC).

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

**Appendix A. Estimation of** $\Pr\{S_i(0) = 1 | X_i = x\}$ **and** $\Pr\{S_i(1) = 1 | X_i = x\}$

We use the maximum likelihood approach to estimate $\Pr\{S_i(0) = 0 | X_i = x\}$, $\Pr\{S_i(0) = 1 | X_i = x\}$ and $\Pr\{S_i(1) = 1 | X_i = x\}$. Let

$$E_{jkx} = \{i : S_i(0) = j, S_i(1) = k | X_i = x\}, \quad j, k = 0, 1, x \in \Gamma$$

be the principal stratum under each category $X = x$. Because of the monotonicity assumption, $E_{10x}$ is empty. Let

$$p_{jkx} = \Pr\{E_{jkx}\} = \Pr\{S_i(0) = j, S_i(1) = k | X_i = x\}, \quad j, k = 0, 1, x \in \Gamma$$

Therefore, $p_{00x} + p_{01x} + p_{11x} = 1$ for all $x \in \Gamma$. For each $x$, $\Pr\{E_{jkx}\}$ can be estimated from the observed data $\{Z_i, X_i, S_i(Z_i), i = 1, 2, \ldots, n\}$ via maximum likelihood. Let $N_{zsx}$ be the total number of subjects with $Z = z$, $S(Z) = s$ and baseline category $x$ with $\sum_{Z;S=0,1;X} N_{zsx} = n$. Then the likelihood function for $(p_{00x}, p_{01x}, p_{11x})$ is given by

$$
\begin{aligned}
L(p_{00x}, p_{01x}, p_{11x} | N_{00x}, N_{01x}, N_{10x}, N_{11x}) &\propto f(N_{zsx}) \\
&\propto \Pr\{S(0) = 0 | X = x\}^{N_{00x}} \cdot \Pr\{S(0) = 1 | X = x\}^{N_{01x}} \\
&\quad \cdot \Pr\{S(1) = 0 | X = x\}^{N_{10x}} \cdot \Pr\{S(1) = 1 | X = x\}^{N_{11x}} \\
&= (p_{00x} + p_{01x})^{N_{00x}} \cdot p_{11x}^{N_{01x}} \cdot p_{00x}^{N_{10x}} \cdot (p_{01x} + p_{11x})^{N_{11x}} \quad \text{(by monotonicity assumption)} \\
&= (1 - p_{11x})^{N_{00x}} \cdot p_{11x}^{N_{01x}} \cdot p_{00x}^{N_{10x}} \cdot (1 - p_{00x})^{N_{11x}} \\
&= (1 - p_{11x})^{N_{00x}} \cdot p_{11x}^{N_{01x}} \cdot (1 - p_{+1x})^{N_{10x}} \cdot p_{+1x}^{N_{11x}}
\end{aligned}
$$

(1) When $N_{00x} \cdot N_{11x} \geq N_{01x} \cdot N_{10x}$, the resulting MLEs for $(p_{00x}, p_{01x}, p_{11x})$ are given by

$$
\begin{aligned}
\widehat{p}_{00x} &= \widehat{\Pr}\{S_i(0) = 0, S_i(1) = 0 | X_i = x\} = 1 - \widehat{p}_{+1x} \\
&= \frac{N_{10x}}{N_{10x} + N_{11x}} = \frac{\sum_i I(Z_i = 1, S_i(1) = 0, X_i = x)}{\sum_i I(Z_i = 1, X_i = x)} \\
\widehat{p}_{11x} &= \widehat{Pr}\{S_i(0) = 1, S_i(1) = 1 | X_i = x\} = \frac{N_{01x}}{N_{00x} + N_{01x}} = \frac{\sum_i I(Z_i = 0, S_i(0) = 1, X_i = x)}{\sum_i I(Z_i = 0, X_i = x)} \\
\widehat{p}_{01x} &= \widehat{Pr}\{S_i(0) = 0, S_i(1) = 1 | X_i = x\} = 1 - \widehat{p}_{00x} - \widehat{p}_{11x}
\end{aligned}
$$

Obviously for each $x \in \Gamma$, $\widehat{p}_{00x}$ is the proportion of non-respondents in the treatment arm with $X = x$; $\widehat{p}_{11x}$ is the proportion of respondents in the control arm with $X = x$.

(2) When $N_{00x} \cdot N_{11x} < N_{01x} \cdot N_{10x}$, $\widehat{p}_{11x} = \widehat{p}_{+1x}$. The likelihood function is given by

$$
\begin{aligned}
L(p_{00x}, p_{01x}, p_{11x} | N_{00x}, N_{01x}, N_{10x}, N_{11x}) \\
= (1 - p_{11x})^{N_{00x}} \cdot p_{11x}^{N_{01x}} \cdot (1 - p_{11x})^{N_{10x}} \cdot p_{11x}^{N_{11x}} \\
= (1 - p_{11x})^{N_{00x} + N_{10x}} \cdot p_{11x}^{N_{01x} + N_{11x}}
\end{aligned}
$$

The resulting MLEs for $(p_{00x}, p_{01x}, p_{11x})$ are given by

$$\widehat{p}_{01x} = \widehat{Pr}\{S_i(0) = 0, S_i(1) = 1 | X_i = x\} = 0$$

$$\widehat{p}_{00x} = \widehat{\Pr}\{S_i(0) = 0, S_i(1) = 0 | X_i = x\} = \frac{N_{+0x}}{N_{++x}} = \frac{\sum_i I(S_i = 0, X_i = x)}{\sum_i I(X_i = x)}$$

$$\widehat{p}_{11x} = \widehat{Pr}\{S_i(0) = 1, S_i(1) = 1 | X_i = x\} = \frac{N_{+1x}}{N_{++x}} = \frac{\sum_i I(S_i = 1, X_i = x)}{\sum_i I(X_i = x)}$$

Then $\widehat{p}_{00x}$ is the proportion of non-respondents among all subjects with $X = x$; $\widehat{p}_{11x}$ is the proportion of respondents among all subjects with $X = x$.

## Appendix B. Proof of Consistency of Model Parameters and Causal Estimands

Here we show our estimator $\widehat{\boldsymbol{\beta}}$ is a consistent estimator for $\boldsymbol{\beta}$. We first show that $\widehat{\boldsymbol{\beta}}$ can be considered as an extremum estimator as defined by Hayashi (2000). Then we prove that the conditions set forth by Hayashi (2000) for consistency of an extremum estimator are satisfied by our estimator. Then by Slutsky's theorem, the causal estimand $\widehat{\theta}$ is a consistent estimator for $\theta$.

**Definition 4 (Extremum Estimator)** *An estimator $\widehat{\eta}$ is an extremum estimator if there is a function $Q_n(\eta)$ such that (Hayashi, 2000)*

$$\widehat{\eta} = \arg\max_{\eta} Q_n(\eta); \ \ \eta \in H.$$

One example of an extremum estimator is the maximum likelihood estimator where

$$Q_n(\eta) = \prod_{i=1}^{n} f(x_i | \eta).$$

Here we minimize the objective function,

$$Q_n(\boldsymbol{\beta}) = \sum_{x=0}^{K} Q_n^{(x)}(\boldsymbol{\beta})$$

$$= \sum_{x=0}^{K} \{\widehat{G}_L(x) - \sum_{y=0}^{1} G_M(x, y; \boldsymbol{\beta}) \cdot \widehat{G}_R(x, y)\}^2; \ x \in \Gamma$$

which is equivalent to maximizing $-Q_n(\boldsymbol{\beta})$. Therefore $\widehat{\boldsymbol{\beta}}$ is an extremum estimator.

Let

$$Q_0(\boldsymbol{\beta}) = \sum_{x=0}^{K} Q_0^{(x)}(\boldsymbol{\beta}); \ x \in \Gamma$$

where $Q_0^{(x)}(\boldsymbol{\beta}) = \{G_L(x) - \sum_{y=0}^{1} G_M(x, y; \boldsymbol{\beta}) \cdot G_R(x, y)\}^2$. We present sufficient conditions for the existence of a unique local minimizer of $Q_0(\boldsymbol{\beta})$ in Lemma 5.

**Lemma 5** *There exists a unique local minimizer $\boldsymbol{\beta_0}$ for $Q_0(\boldsymbol{\beta})$ if:*

*(a)* $Q_0^{(x)}(\boldsymbol{\beta_0}) = 0, \forall x \in \Gamma = \{0, 1, 2, \ldots, K\}$.

*(b)* $\operatorname{rank}\left|\dfrac{\partial \tilde{Q}_0(\boldsymbol{\beta})}{\partial \boldsymbol{\beta}}\right|_{\boldsymbol{\beta}=\boldsymbol{\beta_0}} \geq \dim(\boldsymbol{\beta})$ *where* $\tilde{Q}_0(\boldsymbol{\beta}) = \{Q_0^{(0)}(\boldsymbol{\beta}), Q_0^{(1)}(\boldsymbol{\beta}), \ldots, Q_0^{(K)}(\boldsymbol{\beta})\}^T.$

**Proof** From (a) we have that $\boldsymbol{\beta_0}$ minimizes $Q_0(\boldsymbol{\beta})$ since $Q_0(\boldsymbol{\beta}) \geq 0, \forall \boldsymbol{\beta}$ and $Q_0(\boldsymbol{\beta_0}) = 0$.

Then from (b) and the Implicit Function Theorem, there exists a unique function $g\{\boldsymbol{G_L(x)}, \boldsymbol{G_R(x,y)}\}$ such that $g\{\boldsymbol{G_L(x)}, \boldsymbol{G_R(x,y)}\} = \boldsymbol{\beta_0}$, in the neighborhood of $\{\boldsymbol{G_L(x)}, \boldsymbol{G_R(x,y)}\}$ where $\{\boldsymbol{G_L(x)}, \boldsymbol{G_R(x,y)}\} = [G_L(x), G_R(x,y); x \in \{0, 1, \ldots, K\}, y = 0, 1]$. Thus, $\boldsymbol{\beta_0}$ is a unique local minimizer for $Q_0(\boldsymbol{\beta})$. ∎

The proof of Theorem 3 is given as below.

**Proof** From Proposition 7.1 in (Hayashi, 2000): an extremum estimator $\widehat{\eta}$ is a consistent estimator for $\eta$ if there is a function $Q_0(\eta)$ satisfying the following two conditions:

(I) Identification: $Q_0(\eta)$ is uniquely maximized on $H$ at $\eta_0 \in H$.

(II) Uniform convergence: $Q_n(\cdot)$ converges uniformly in probability to $Q_0(\cdot)$.

The condition (I) is satisfied according to Lemma 5. To show that the condition (II) is satisfied here, let

$$
\begin{aligned}
Q_n(\boldsymbol{\beta}) &= \sum_{x=0}^{K} Q_n^{(x)}(\boldsymbol{\beta})^2 \\
&= \sum_{x=0}^{K} \{\widehat{G}_L(x) - \sum_{y=0}^{1} G_M(x,y;\boldsymbol{\beta}) \cdot \widehat{G}_R(x,y)\}^2 \\
Q_0(\boldsymbol{\beta}) &= \sum_{x=0}^{K} Q_0^{(x)}(\boldsymbol{\beta})^2 \\
&= \sum_{x=0}^{K} \{G_L(x) - \sum_{y=0}^{1} G_M(x,y;\boldsymbol{\beta}) \cdot G_R(x,y)\}^2.
\end{aligned}
$$

From

$$
\begin{aligned}
|Q_n(\boldsymbol{\beta}) - Q_0(\boldsymbol{\beta})| &= |\sum_{x=0}^{K} Q_n^{(x)}(\boldsymbol{\beta})^2 - \sum_{x=0}^{K} Q_0^{(x)}(\boldsymbol{\beta})^2| \\
&\leq \sum_{x=0}^{K} |Q_n^{(x)}(\boldsymbol{\beta})^2 - Q_0^{(x)}(\boldsymbol{\beta})^2| \\
&= \sum_{x=0}^{K} |Q_n^{(x)}(\boldsymbol{\beta}) - Q_0^{(x)}(\boldsymbol{\beta})| \cdot |Q_n^{(x)}(\boldsymbol{\beta}) + Q_0^{(x)}(\boldsymbol{\beta})| \\
&\leq \sum_{x=0}^{K} 2 \cdot |Q_n^{(x)}(\boldsymbol{\beta}) - Q_0^{(x)}(\boldsymbol{\beta})|, \quad x \in \Gamma
\end{aligned}
$$

because $0 \leq |Q_n^{(x)}(\boldsymbol{\beta})| \leq 1$ and $0 \leq |Q_0^{(x)}(\boldsymbol{\beta})| \leq 1$, each of which is a difference of two probability estimates.

Therefore,

$$|Q_n(\boldsymbol{\beta}) - Q_0(\boldsymbol{\beta})|$$

$$\leq \sum_{x=0}^{K} 2 \cdot \left\{ |\widehat{G}_L(x) - G_L(x)| + \sum_{y=0}^{1} G_M(x, y; \boldsymbol{\beta}) \cdot |\widehat{G}_R(x, y) - G_R(x, y)| \right\}$$

$$\leq \sum_{x=0}^{K} 2 \cdot \left\{ |\widehat{G}_L(x) - G_L(x)| + \sum_{y=0}^{1} |\widehat{G}_R(x, y) - G_R(x, y)| \right\} \tag{8}$$

because $G_M(x, y; \boldsymbol{\beta})$ is a probability bounded between 0 and 1.

Since $\widehat{G}_L(x)$ and $\widehat{G}_R(x, y)$ are either sample proportions or their ratios,

$$\widehat{G}_L(x) \xrightarrow{p} G_L(x), \text{ as } n \to \infty$$

$$\widehat{G}_R(x, y) \xrightarrow{p} G_R(x, y), \text{ as } n \to \infty$$

As $\widehat{G}_L(x)$ and $\widehat{G}_R(x, y)$ do not involve $\boldsymbol{\beta}$, from (8) we have

$$Q_n(\boldsymbol{\beta}) \overset{p}{\Longrightarrow} Q_0(\boldsymbol{\beta}), \text{ as } n \to \infty$$

where $\overset{p}{\Longrightarrow}$ denotes uniform convergence in probability. This confirms condition (II) and completes the proof of $\widehat{\boldsymbol{\beta}} \xrightarrow{p} \boldsymbol{\beta}$ as $n \to \infty$.

Because the causal estimate $\widehat{\theta}$ is a continuously differentiable function of $\widehat{\boldsymbol{\beta}}$ and relevant sample proportions, by Slutsky's theorem, $\widehat{\theta} \xrightarrow{p} \theta$ as $n \to \infty$. ∎

## Appendix C.  Calculation of True Principal Stratum Causal Effects

For the simulated data, the true average causal effect for principal stratum $S_i(1) = 1$ can be calculated by

$$\mathbb{E}\{Y_i(1) - Y_i(0)|S_i(1) = 1\} = \mathbb{E}\{Y_i(1) = 1|S_i(1) = 1\} - \mathbb{E}\{Y_i(0) = 1|S_i(1) = 1\}$$
$$= \frac{\Pr\{Y_i(1) = 1, S_i(1) = 1\} - \Pr\{Y_i(0) = 1, S_i(1) = 1\}}{\Pr\{S_i(1) = 1\}}$$

where

$$\Pr\{S_i(1) = 1\} = \sum_{x} \Big\{ \Pr\{S_i(0) = 1|X_i = x\} \cdot \Pr\{X_i = x\}$$
$$+ \sum_{y} \big[ \Pr\{X_i = x\} \cdot \Pr\{S_i(0) = 0|X_i = x\}$$
$$\cdot \Pr\{Y_i(0) = y|S_i(0) = 0, X_i = x\}$$
$$\cdot \Pr\{S_i(1) = 1|S_i(0) = 0, Y_i(0) = y, X_i = x\} \big] \Big\}$$

$$\Pr\{Y_i(0) = 1, S_i(1) = 1\} = \sum_x \Big[ \Pr\{X_i = x\} \cdot \Pr\{S_i(0) = 1 | X_i = x\}$$

$$\cdot \Pr\{Y_i(0) = 1 | S_i(0) = 1, X_i = x\}$$
$$+ \Pr\{X_i = x\} \cdot \Pr\{S_i(0) = 0 | X_i = x\}$$
$$\cdot \Pr\{Y_i(0) = 1 | S_i(0) = 0, X_i = x\}$$
$$\cdot \Pr\{S_i(1) = 1 | S_i(0) = 0, Y_i(0) = 1, X_i = x\} \Big]$$

$$\Pr\{Y_i(1) = 1, S_i(1) = 1\} = \sum_x \sum_y \Big[ \Pr\{X_i = x\} \cdot \Pr\{S_i(0) = 1 | X_i = x\}$$

$$\cdot \Pr\{Y_i(0) = y | S_i(0) = 1, X_i = x\}$$
$$\cdot \Pr\{Y_i(1) = 1 | Y_i(0) = y, S_i(0) = 1, X_i = x\}$$
$$+ \Pr\{X_i = x\} \cdot \Pr\{S_i(0) = 0 | X_i = x\}$$
$$\cdot \Pr\{Y_i(0) = y | S_i(0) = 0, X_i = x\}$$
$$\cdot \Pr\{S_i(1) = 1 | S_i(0) = 0, Y_i(0) = y, X_i = x\}$$
$$\cdot \Pr\{Y_i(1) = 1 | S_i(0) = 0, S_i(1) = 1, Y_i(0) = y, X_i = x\} \Big]$$

