# OpenReview forum: "Identifying Principal Stratum Causal Effects Conditional on a Post-treatment Intermediate Response"
_cclear.cc/CLeaR/2022/Conference — CLeaR 2022 Poster_

### Official Review · Reviewer_veHz · 2021-11-16

**Confidence:** 4
**Overall Score:** 7

**Main Review:**

I believe the theorems and propositions in this article are interesting in their own right.  The technical results appear to be correct.  But it seems that this method is used to estimate the causal effect in $E_{01} \cup E_{11}$ instead of $E_{01}$ only.  Why not use principal strata
{S(1)=1} and {S(0)=0}?  Using four principal strata seems to complicate this problem.
I might improve my score after my questions are well-answered.

I am sorry. Because $P(Y(0)| S(1)=1)$ is cross-world, it seems that the principal stratification framework ( 4  principal strata) and monotonicity assumption are necessary.

**Summary:**

Under mild assumptions, this propose an approach to estimate those model parameters using empirical data and subsequently the causal estimand of interest.  And  apply this method to a recent clinical trial and its performance is evaluated via simulation studies. The main contribution of this paper is to propose a new method to identify and estimate principal stratum causal effects under  new data settings.

---

> ### Author Response · Authors · 2021-11-30
> **Author Response to Reviewer veHz**
>
> We thank the reviewer for reviewing the paper and providing insightful feedback. In the paper, motivated by our real-data application, the NSABP B-40 study, we are interested in the causal effect in $E_{+1} = E_{01} \cup E_{11}$ for those who would achieve pCR had they been treated with chemotherapy plus bevacizumab. Other principal stratum causal effects such as $\theta_{jk}$ can be estimated using a similar approach as we outline in Section 4. For example, to estimate the causal effect in $E_{01}$ requires conditioning on $S(0)=0$ and $S(1)=1$ as in equations (4) and (5). All relevant terms can be expanded as in equation (6). Regression models for counterfactual outcomes, $Pr[S(1)=1|S(0)=0, Y(0)=y,X=x]$, and $Pr[S(0)=1|S(1)=1, Y(1)=y, X=x]$ need to be specified. The regression coefficients can be identified and estimated according to the proposed method. If the paper is accepted, to properly elucidate the proposed method, we’ll add a subsection in Section 4 to present the main results on the identification of regression coefficients of the model for counterfactual outcomes, as in equation (1). You are correct that monotonicity is a key assumption in order to achieve parameter identifiability as in many causal inference methods. This assumption is popular and can be justified in many real data.

---

### Official Review · Reviewer_M4f1 · 2021-11-23

**Confidence:** 4
**Overall Score:** 7

**Main Review:**

High level overview of comments:
-	This is a very interesting, well-motivated problem and I enjoyed reading your paper. There are only two areas where I think would significantly improve your paper if you could address. They are listed in the below two bullets.
-	Clarity of assumptions and notation:  You mention that identification is tricky for principal stratum causal effects, but you do not explicitly say anywhere that the assumptions you make are sufficient for identification. Additionally, it would be helpful to explicitly define the \beta used in section 4.2.
-	Clarifying methodological contribution: Of course, you have a contribution in terms of the application to breast cancer trials, but it’s not entirely clear from reading (especially the related work section and the introduction) whether you are trying to say that you have a methodological contribution in this paper as well. You say at the end of section 2 that you propose a method to identify and estimate principal stratum effects under a setting similar to Shepherd et al, but for binary outcomes. What exactly is Shepherd et al. doing? How significant of an extension is your work?


More detailed comments:
-	I am wondering if your problem could be approached using a mediational analysis framework, where pCR would be a looked as a mediator for survival. Could you discuss why this would or would not make sense?
-	I would be nice if either before or within your related work section you mathematically write out what a principal stratum causal effect is (you write this nicely in section 3.1, but it would be easier for someone who is not familiar with principal stratum, but is familiar with potential outcomes, to understand high level what you are doing).
-	Additionally, it would be helpful to clarify exactly what your contribution is. You say at the end of section 2 that you propose a method to identify and estimate principal stratum effects under a setting similar to Shepherd et al. Could you clarify what Shepherd does differently / if that method is applicable to binary outcomes?
-	Before you introduce the estimators you will use in 4.1, I think it would be helpful to formally write out the assumptions needed for identification of the causal estimand (e.g. a theorem proving identifiability using the stated assumptions). It is not clear to me from reading if the assumptions you describe in section 3 are sufficient for identification. It would especially help readers not familiar with principal stratification who are unfamiliar with what kind of assumptions are needed for identification, especially since you say in section 2 that identification of principal stratum causal effects is generally difficult!
-	For Theorem 3, it would be helpful if you list/refer to explicitly all the assumptions you are using (e.g., treatment assignment is independent of potential outcomes, monotonicity of outcomes S_i). Moreover, it would be helpful to discuss what exact
-	In section 4.2, you define G_M(x,y,; \beta). I think this is the same beta discussed in section 3.2. It would be helpful to explicitly define \beta here definition. How many dimensions in \beta here?
-	In section 6.3 you say that you compare the performance of your method with the sensitivity analysis similar to Gilbert et al and Shephard et al. I am confused exactly with what you are trying to say here. Are you saying you are (i) performing sensitivity analysis of your method with a technique similar to that of Gilbert et al and Shephard et al OR (ii) are you saying that there is a completely different way to approach this problem / different estimator that you are comparing to. Based on the first sentence in 6.3 it sounds like case (ii), but based on what you write in the rest of 6.3 it sounds like case (i).


**Summary:**

This paper works on analyzing the effect of treatment on survival among people who have a positive intermediate binary outcome in neoadjuvant breast cancer clinical trials. They use a principal stratum framework to approach the problem and propose an estimator for the causal effects of interests, specifically for binary outcomes.

---

> ### Author Response · Authors · 2021-11-30
> **Author Response to Reviewer M4f1**
>
> We thank the reviewer for reviewing the paper and providing insightful feedback. Below are our clarifications to the questions and comments.
>
> Regarding your high-level overview of comments:
> - Even though we discussed the identification of regression coefficients in a couple of places (bottom of page 6 in Section 4.2 and Section 7), this issue is essential to this paper. We will add a sub-section on the identifiability in Section 4.
> - We adopted the same logistic regression model and assumptions as in Shepherd et al. (2006). In general, regression coefficients are not identifiable without any additional assumptions. However, In that paper and subsequent works, Shepherd and his co-authors their work focused on sensitivity analyses by varying model parameters in an imposed model on counterfactual outcomes. Noticing the probabilistic equation (7), we figured out how to identify the regression coefficients when the linearity assumption holds in the logistic regression. Indeed we developed a new method for the identification and estimation of regression coefficients and subsequent causal estimands.
> - Although we discussed the required conditions/assumptions (a. monotonicity; b. the number of levels for X is greater than or equal to 2) for identification of regression coefficients in a couple of places (bottom of page 6 in Section 4.2 and Section 7), the presentation of this important issue needs to be improved. We will add a sub-section on the identifiability in Section 4.
>
> Regarding your detailed comments:
> - Under the principal stratification framework, our casual estimands of interest in section 3 cannot be addressed via mediation analysis with pCR as a mediator.
> - The mathematical characterization of principal stratum causal effects is presented in Section 3.1. We will provide a couple of sentences to illustrate its interpretation in the revision.
> - Indeed we provided a method to identify and estimate regression coefficients for the counterfactual model which nobody was able to accomplish in the past. We will clarify our contribution in the revision.
> - We discussed the two required conditions in Section 4.2 and Section 7. We agree that the presentation is inadequate and will add a new subsection on the identification issue in Section 4 because it is essential to this manuscript.
> - We will add the exact conditions required for estimation in the revision.
> - The $\beta$ in Section 3.2 and Section 4.2 are the same. The $\beta$ is of dimension 3, including the intercept, the coefficient for $Y_i(0)$, and the coefficient for the baseline covariate $x$.
> - Unable to identify regression coefficients, Shepherd et al. conducted a sensitivity analysis for such data. Here we proposed a new method to identify and estimate regression coefficients. Comparing sensitivity analysis results to our method provides the reader the context that estimates from sensitivity analysis may vary a lot without taking advantage of what is presented by the observed data.

---

### Official Review · Reviewer_Lefw · 2021-11-23

**Confidence:** 3
**Overall Score:** 7

**Main Review:**

Pros:
1.	The article is fairly well written and precise.
2.	Identification of principal stratum effects is an important topic.
3.	The article has a nice application to real data.

Cons/issues:
1.	I had a bit of a difficult time understanding the novel element of this article and what distinguished it from Shepherd et al. (2006). I am confused by the logistic regression model, which along with monotonicity appears to be sufficient to identify the principal stratum effect of interest. However, my understanding is that Shepherd et al. (2006) used this model for sensitivity analysis, suggesting that this effect was not identified. So did they in fact not need to vary the $\beta$ coefficients in this model because they were already identified? Or were they operating under a different set of assumptions in which this model and the principal stratum effect were not identified? It would help to clear up this confusion if the authors could clarify the distinction between their approach and that of Shepherd et al. (2006).
2.	The main challenge taken on in this article seems to be about identification; however, the article seemed to skip over identification and mainly talk about estimation. The article would be much easier to read in my opinion if there were a sharper division between identification and estimation. That is, it would be much more enlightening to have identified quantities and estimators all written in terms of observed variables rather than potential outcomes. This might also help to address my previous point.
3.	Relatedly, it is not very clear how the joint distribution of $S(0)$ and $S(1)$ in $G_L$ is identified, and the role the monotonicity assumption plays in this.
4.	I think it would make more sense to move Section 3.2 after Section 4.1.
5.	I did not understand very well what role $X$ plays. Is it just for precision? It seems you still have identification even without it.
6.	Does $Q_0^{(x)}$ depend on $y$? This is not reflected in the notation.
7.	Do the $\beta_x$s in Section 6.3 correspond to a larger model with dummy variables for each level of $x$ rather than assuming linearity in $x$?

Shepherd, B. E., Gilbert, P. B., Jemiai, Y., & Rotnitzky, A. (2006). Sensitivity analyses comparing outcomes only existing in a subset selected post‐randomization, conditional on covariates, with application to HIV vaccine trials. Biometrics, 62(2), 332-342.


**Summary:**

This article proposes an approach to identifying and estimating a principal stratum effect based on a monotonicity assumption and a logistic regression model on a set of observed and counterfactual variables whose joint distribution is not nonparametrically identifiable on its own, as was proposed by Shepherd et al. (2006) for sensitivity analysis. The authors then apply this methodology to estimate principal stratum effects of bevacizumab as a supplement to neoadjuvant chemotherapy on event-free survival and overall survival.

---

> ### Author Response · Authors · 2021-11-30
> **Author Response to Reviewer Lefw**
>
> We thank the reviewer for reviewing the paper and providing insightful feedback. In the following, we provide responses and clarification on issues raised by the reviewer.
> 1. We adopted the same logistic regression model and assumptions as in Shepherd et al. (2006). In general, regression coefficients are not identifiable without any additional assumptions. In that paper and subsequent works, Shepherd and his co-authors focused on sensitivity analyses by varying model parameters in an imposed model on counterfactual outcomes. Noticing the probabilistic equation (7), we figured out how to identify the regression coefficients when the linearity assumption holds in the logistic regression. The identification was discussed in the last paragraph of page 6 in Section 4.2 and required assumptions were discussed in Section 7. We will revise the paper by providing a sub-section on the identifiability in Section 4.
> 2. We agree with the reviewer that a sub-section on identifiability in Section 4 is warranted. We will revise the manuscript accordingly.
> 3. The joint distribution of $S(0)$ and $S(1)$ in $G_L$ can only be estimated with maximum likelihood from the observed data under the monotonicity assumption. Details are presented in Lemma 1 and Appendix A. We will clarify its importance after Lemma 1 in the revision.
> 4. We agree it would help with the flow if we move section 3.2 after section 4.1. However, imposing regression models (1) and (2) among counterfactuals is critical for developing the relevant causal estimands and we did not invent the wheel here. Having section 3.2 in its current place offers us an opportunity to give other researchers proper credits for their contribution.
> 5. Having the baseline covariate $X$ is essential to the identification of the principal stratum causal effects from the observed data. Its relevance to identifiability was briefly discussed in the last paragraph of section 4.2 on page 6. We will clarify in the additional sub-section on identification in the revision.
> 6. No, $Q_0^{(x)}$ does not depend on $y$. There is a summation on $y$.
> 7. Yes, the regression model would become $G_M(x,y;{\beta}) = \displaystyle{\frac{\exp(\beta_x+\beta_{1}y)}{1+\exp(\beta_x+\beta_{1}y)}}$. No linearity assumption is necessary once $\beta_1$ is assumed known. For each X=x, the parameter $\beta_x$ can be exactly identified and estimated.

---

> > ### Comment · Reviewer_Lefw · 2021-12-28
> > **Reviewer Response**
> >
> > Thank you very much for your detailed reply. I must confess that I still do not quite understand what assumptions you are invoking beyond those of Shepherd et al. Your assumptions must be stronger than those of Shepherd et al. since they do not obtain point identification, but from the article and your response it remains unclear to me how they differ. Based on Section 6.3, it seems that perhaps Shepherd et al. used a logistic model with dummy variables for each level of X, rather than a model with a single regression coefficient corresponding to X, i.e., assuming linearity in X. Is this the case? If so, the logistic model (1) ought to be modified. In any case, I hope that this ambiguity can be resolved or clarified in the new subsection on identifiability.

---

### Decision · Program_Chairs · 2022-01-13

**Decision:**

Accept (Poster)

**Comment:**

The authors propose a method to estimate the principal stratum effect of a treatment on the survival outcome of patients with a positive intermediate binary outcome in neoadjuvant breast cancer clinical trials. The assumptions of the paper are similar to that of Shepherd et al. (2006) and include monotonicity and a logistic regression model. The authors include theoretical as well as applied results. This is a well-written paper and the work within is important and clearly impactful. All reviewers see merit in the publication of this article.

However, as promised by the authors in the discussion, the final version of the article should address the following two points: (1) add a section about the identification assumptions (2) clarify their contribution compared to Shepherd et al. (2006).